# CHARFORMER: FAST CHARACTER TRANSFORMERS VIA GRADIENT-BASED SUBWORD TOKENIZATION

**Yi Tay**[*], **Vinh Q. Tran**[*], **Sebastian Ruder**[†], **Jai Gupta, Hyung Won Chung, Dara Bahri**
**Zhen Qin, Simon Baumgartner, Cong Yu, Donald Metzler**
Google Research and DeepMind[†]
yitay@google.com, vqtran@google.com

## ABSTRACT

State-of-the-art models in natural language processing rely on separate rigid subword tokenization algorithms, which limit their generalization ability and adaptation to new settings. In this paper, we propose a new model inductive bias that learns a subword tokenization end-to-end as part of the model. To this end, we introduce a soft gradient-based subword tokenization module (GBST) that automatically learns latent subword representations from characters in a data-driven fashion. Concretely, GBST enumerates candidate subword blocks and learns to score them in a position-wise fashion using a block scoring network. We additionally introduce CHARFORMER, a deep Transformer model that integrates GBST and operates on the byte level. Via extensive experiments on English GLUE, multilingual, and noisy text datasets, we show that CHARFORMER outperforms a series of competitive byte-level baselines while generally performing on par and sometimes outperforming subword-based models. Additionally, CHARFORMER is fast, improving the speed of both vanilla byte-level and subword-level Transformers by 28-100% while maintaining competitive quality. We believe this work paves the way for highly performant token-free models that are trained completely end-to-end.

## 1 INTRODUCTION

Neural networks have achieved tremendous success in natural language processing (NLP) by replacing feature-engineered models with stacks of functions that are learned end-to-end from vast amounts of data (Mikolov et al., 2013; Peters et al., 2018; Howard and Ruder, 2018). The single component of the traditional NLP pipeline (Manning and Schütze, 1999) that has so far resisted gradient-based learning is tokenization, which is commonly applied as a pre-processing step. State-of-the-art pre-trained language models (Devlin et al., 2019) generally rely on data-driven subword-based tokenization algorithms (Schuster and Nakajima, 2012; Sennrich et al., 2016; Wu et al., 2016; Kudo and Richardson, 2018) while expert-crafted segmentation algorithms are still common for languages without whitespace separation such as Chinese, Thai, and Korean (cf. Lample and Conneau, 2019).

This reliance on rigid tokenization methods introduces a bottleneck into current NLP systems that limits their capabilities. Subword segmentation algorithms split tokens into subwords solely based on frequency, without taking into account lexical or semantic similarity. As a result, models are brittle to rare words (Gong et al., 2018) and perturbations, both natural and adversarial (Belinkov and Bisk, 2018; Pruthi et al., 2019; Sun et al., 2020). In multilingual models, tokens in low-resource languages are split into many subwords, which impacts performance on those languages and deteriorates cross-lingual transfer (Hu et al., 2020; Wang et al., 2021). Finally, a separate tokenization algorithm leads to a mismatch between the pre-training and downstream distribution of words when adapting pre-trained language models to new settings, which requires significant engineering effort to overcome.

The direct application of character-level modelling into pre-trained language models in turn results in severely increased computational and memory complexity due to an increased sequence length and generally lower performance. To address this problem, we propose gradient-based subword tokenization (GBST), a new method that combines the compositionality of character-level representations

---

[*]Equal Contribution

with the efficiency of subword tokenization while enabling end-to-end learning. Our method learns latent subword representations from characters using large amounts of unlabeled data. Specifically, GBST learns a position-wise soft selection over candidate subword blocks by scoring them with a scoring network. In contrast to prior tokenization-free methods (Clark et al., 2021), GBST learns interpretable latent subwords, which enables easy inspection of lexical representations and is more efficient than other byte-based models (Xue et al., 2021). Given that simply applying a standard Transformer on a sequence of characters and bytes is computationally prohibitive, GBST paves the way for usable, practical and highly performant character-level models. A high level overview of how the GBST module is applied can be found at Figure 3 (Appendix).

We furthermore introduce CHARFORMER, a Transformer encoder-decoder model that uses GBST to operate directly on the byte level. In addition, we experiment with a re-scaled variant of CHAR-FORMER, which allocates additional capacity to the encoder to make up for the lack of discrete subword embeddings.

We evaluate our model on a range of standard and non-standard English, and multilingual downstream tasks. On English GLUE and long document classification tasks, CHARFORMER outperforms strong byte-level baselines and overall achieves performance on par with subword-based models such as BERT (Devlin et al., 2019) and T5 (Raffel et al., 2020). On toxicity detection in social media datasets (Borkan et al., 2019; Wulczyn et al., 2017), CHARFORMER outperforms byte-level baselines as well as subword-based models, demonstrating robustness to spelling variation and non-standard language. Finally, a multilingually pre-trained CHARFORMER performs on par or outperforms strong subword-based multilingual baselines on standard cross-lingual datasets.

We additionally demonstrate CHARFORMER is more efficient compared to byte-level and subword-based models with similar numbers of parameters. On a comparable setup, CHARFORMER outperforms a baseline similar to the recent state-of-the-art byte-level model ByT5 (Xue et al., 2021) while being $2\times$ more memory efficient and 10–93% faster. CHARFORMER also trains 28% faster than the subword-level mT5 model (Xue et al., 2020), has $3\times$ fewer parameters and achieves comparable quality on well-established benchmarks. Finally, we demonstrate via visualization that the latent subwords learned by CHARFORMER are interpretable to some extent.

## 2 CHARFORMER

This section introduces our efficient character-level architecture, CHARFORMER. CHARFORMER is comprised of a Gradient-Based Subword Tokenization (GBST) module, followed by deep Transformer layers. The input to the GBST module is a sequence of characters or bytes[1], which is then downsampled to construct *latent subwords*.

### 2.1 GRADIENT-BASED SUBWORD TOKENIZATION (GBST)

The input to GBST is a tensor of shape $X \in \mathbb{R}^{L \times d}$ where $L$ is the number of input characters and $d$ is the character embedding dimension. The key idea behind GBST is for the model to learn to perform a latent subword segmentation of the input by selecting the most suitable subword block at every character position. A block is a contiguous span of characters $X_{i:i+b}$ of length $b$ for $1 \le i \le L - b$.

#### 2.1.1 CONSTRUCTING CANDIDATE LATENT SUBWORD BLOCKS

We first enumerate all possible subword blocks of size $b$ up to a maximum block size $M$. In order to learn subword block embeddings, we use a non-parameterized strided pooling function $F : \mathbb{R}^{b \times d} \to \mathbb{R}^d$ that projects a subword block consisting of a sequence of character embeddings $X_{i:i+b} \in \mathbb{R}^{b \times d}$ to a single subword block representation $X_{b,i} \in \mathbb{R}^d$ for block size $b$ at position $i$. We compute subword blocks $X_{b,i}$ with a stride $s$:

$$X_b = [F(X_{i:i+b}); F(X_{(i+s):(i+s)+b}); \ldots] \tag{1}$$

---

[1]We choose bytes rather than characters (Unicode code points) as this allows us to use a vocabulary of 256 possible byte values for all settings. We note that for languages with a Latin alphabet, many characters correspond to a single byte. For other languages, each character corresponds to 2–3 bytes in general. For simplicity and to align with prior work, we will generally talk about characters unless stated otherwise.

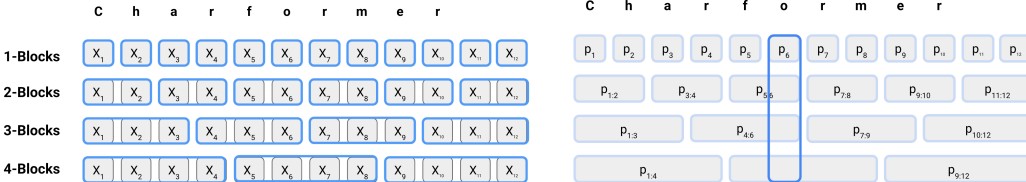

(a) Formation of subword blocks to be scored by $F_R$. Offsets and/or pre-GBST convolutions not shown.

(b) Block scores that have been expanded back to length $L$. Softmax is taken over block scores at each position $i$ to form block weights for constructing latent subword representations.

Figure 1: Illustration of subword block formation and scoring.

In practice we set $s = b$, thus $X_b \in \mathbb{R}^{\frac{L}{b} \times d}$. The construction of latent subword blocks creates a shorter overall sequence length by downsampling. We construct $X_b$ for $b \in 1, \dots, M$, which can be seen in Figure 1 for $M = 4$.

**Considering Offsets** A limitation of a strided implementation is that it is unable to model all possible subword windows. For instance, for the character sequence $[a, b, c, d]$ we would only be able to allocate $[a, b]$ and $[c, d]$ as subword blocks of length $b = 2$ and would ignore the subword block $[b, c]$. Offsets can be used to model sliding windows of all possible subword blocks. We consider enumerating all possible strided blocks by additionally shifting sequences up until the offset $s$. As this increases computation, we instead propose to first apply a 1D convolution to $X$, prior to enumerating subword blocks. This effectively "smoothes" over the subword blocks. We use the variant with 1D convolutions in our main experiments and provide additional ablations in §8.3 of the Appendix.

**Considering Intra-block Positions** It is important to preserve the ordering of the characters within the block $X_i, X_{i+1}, \dots, X_{i+b}$. E.g., the output of $F$ should differ for the blocks $abc$ and $bca$. For certain choices of $F$ it may be valuable to add a positional embedding (Vaswani et al., 2017) to $X_{i:i+b}$ before applying $F$. Note that this positional embedding would only be for individual blocks, and is not global to the entire input sequence. That is, only positional embedding values for positions $1, \dots, b$ would be used. However, in practice we apply a 1D convolution before the GBST layer and use the mean-pooling function for $F$. We find this to be sufficient to distinguish between same sized blocks with different character orders.

### 2.1.2    BLOCK SCORING NETWORK

In order to allow the model to learn which block to select for every character position, we introduce a block scoring network. The block scoring network is simply a parameterized function $F_R(.)$ that produces a score for each candidate block. Given a subword candidate block $X_{b,i} \in \mathbb{R}^d$, we compute a score $p_{b,i}$ associated with the block using a simple linear transformation $F_R : \mathbb{R}^d \to \mathbb{R}$:

$$p_{b,i} = F_R(X_{b,i}) \tag{2}$$

We perform ranking of subword blocks with regard to each character position in the original sequence. At every position $i$, the model learns to select the most suitable subword block $X_{b,i}$ among all block sizes $1 \le b \le M$. As each sequence of subword blocks $X_b$ is downsampled, we realign the representations of the subword blocks by upsampling each $X_b$ to its original sequence length $L$. Specifically, for a block size of $b$, we replicate each block representation $X_{b,i}$ $b$ times. We then score each candidate block at each position $i$ using the softmax function:

$$P_i = \text{softmax}([p_{1,i}, p_{1,i}, \cdots, p_{M,i}]), \tag{3}$$

which computes a relative score of each candidate block at each position and $P_i \in \mathbb{R}^M$. We show the scoring of realigned blocks in Figure 1.

### 2.1.3 Forming Latent Subwords

We then sum the representations of all subword blocks $X_{b,i}$ at each position $i$ multiplied by their learned probability $P_{b,i}$ to form a latent subword representation $\hat{X}_i \in \mathbb{R}^d$:

$$\hat{X}_i = \sum_b^M P_{b,i} X_{b,i} \tag{4}$$

Intuitively, the model learns an ideal subword block for each position. In contrast to standard deterministic subword tokenization algorithms, this selection is *soft* and can thus consider different possible segmentations at every position $i$. In general, however, this formulation still assumes that subwords are contiguous sequences of characters. While additional context can be considered via the convolutions in §2.1.1, non-concatenative morphology where morphemes are discontinuous may be harder for the method to model.[2]

### 2.1.4 Position-wise Score Calibration

In the above approach, the scoring of each position is independent of other positions. We hypothesize that it may be beneficial for block scores at each position to be aware of each other. To this end, we introduce an optional module that enables learning a consensus among block scores by calculating dot products across the scores $P_i$ across all positions $i \in [1, L]$. This can be viewed as a form of self-attention across block scores, albeit without any projections for computational efficiency. To learn the new scores $\hat{P} \in \mathbb{R}^{L \times M}$, we compute $\hat{P} = \mathrm{softmax}(PP^\top)P$.

### 2.1.5 Downsampling

After learning a candidate block or mixture of blocks for each position, we use a downsampling function $F_D : \mathbb{R}^{L \times d} \to \mathbb{R}^{\frac{L}{d_s} \times d}$ that downsamples the sequence of latent subwords $\hat{X} = [\hat{X}_1, \ldots, \hat{X}_L]$ to $\tilde{X}$, reducing its sequence length by a factor of $d_s$. We choose $F_D$ to be a non-parameterized mean pooling operation. Notably, such simple stride-based pooling removes potential redundancies caused by adjacent positions selecting similar blocks as the mean pool of two identical block embeddings produces the same outcome. Intuitively, as the downsampling operation is fixed, the parameterized components preceding it should learn an optimal subword tokenization given the downsampling.

## 2.2 Transformer Stack

The remainder of the CHARFORMER model remains identical to a regular Transformer encoder-decoder model. The Transformer stack operates on the downsampled latent subwords $\tilde{X}$ instead of subword embeddings.

**Re-scaling of the Transformer Stack** While subword-based models allocate much of their capacity to subword embeddings—up to 71% of all parameters for contemporary multilingual models (Chung et al., 2021)—, the character vocabulary of character-level models is much smaller and thus less expressive. Similar to Xue et al. (2021), we hypothesize that character-level models require deeper encoder stacks than subword-based models to make up for their smaller embedding capacity. Consequently, we explore a scaling variant of CHARFORMER that puts more parameters at the encoder at the expense of the decoder while preferring a deep narrow model over a larger wide model. Specifically, we re-configure the Base model size to be similar to the T5 Small model size, with an expanded $24$ layers in the encoder. The resulting CHARFORMER$_{SBase}$ (Scaled Base) has $134M$ parameters, which is about 67% the parameter footprint of the standard base T5 model (200M parameters; Raffel et al., 2020). Moreover, this particular CHARFORMER model is approximately 50-100% faster than the T5 base model (see §4).[3] For the re-scaled variant, we also used the GLU variant described in (Shazeer, 2020) which is commonly referred to as the V1.1 variant in the T5 library.

---

[2]Future work could explicitly seek to model discontinuous morphological processes by considering skip-grams in addition to character n-grams, although this would increase computational costs.

[3]The benefits of such re-scaling have also been observed for subword-based encoder-decoder neural machine translation models (Devlin, 2017; Kasai et al., 2021).

**A Note on Comparing Character-level and Subword-based Methods**  Prior work on efficient methods generally compares models with the same number of parameters (Chung et al., 2021). However, whereas embedding look-up even with large vocabularies in subword-based methods is $\mathcal{O}(1)$, re-distributing the subword embedding parameters in character-level models such as ByT5 (Xue et al., 2021) to dense layers incurs much higher computational costs—a 25% penalty in training speed. We believe that a fair re-scaling of character-level models should not only aim to match the number of parameters but also the compute and inference costs of subword-based models under the assumption that char/byte-level models will require longer sequences (see §4 for a comparison).

**Span-based Pre-training**  Our pre-training scheme follows T5 quite closely. We mask $N$ contiguous characters and train to predict them in a sequence-to-sequence architecture following Xue et al. (2021). The model optimizes the cross-entropy loss and is trained with teacher forcing.

## 3  EXPERIMENTS

We evaluate our method both in English as well as in a multilingual setting on relevant benchmarks and compare against state-of-the-art character-level and subword-based methods.

### 3.1  EXPERIMENTS ON MONOLINGUAL ENGLISH DATASETS

**Data**  To showcase the effectiveness of the proposed method, we evaluate on a diverse set of standard English tasks from GLUE covering sentiment classification (SST-2; Socher et al., 2013), natural language inference (MNLI, QNLI; Williams et al., 2018; Rajpurkar et al., 2016), paraphrase detection (Dolan and Brockett, 2005, MRPC, QQP) and sentence similarity (Cer et al., 2017). In addition, we evaluate on tasks that require dealing with long documents, both for sentiment analysis (IMDb; Maas et al., 2011) and news classification (AGNews; Zhang et al., 2015).

**Baselines**  We compare CHARFORMER against the following state-of-the-art subword-based models: BERT (Devlin et al., 2019), an encoder-only pre-trained masked language model; and T5 (Raffel et al., 2020), an encoder-decoder model. We also compare against Byte-level T5 (Xue et al., 2021), a T5 model that is directly applied to bytes. We additionally evaluate the impact of the downsampling in CHARFORMER by comparing it to the downsampling used by the character-level CANINE (Clark et al., 2021) model in our framework. CANINE downsamples a character sequence using local attention and pooling via strided convolutions. As the original CANINE uses an encoder-only model and was only trained on multilingual data, we integrate CANINE-style downsampling into Byte-level T5, which we refer to as Byte-level T5+LASC (local attention–strided convolution).[4] As an ablation for the GBST inductive bias, we compare against Byte-level T5+Conv$_{Base}$ a convolutional baseline of Byte-level T5 with a 1D convolution of filter size 5 placed before the encoder. Note that in all the baselines and for CHARFORMER base models, in the spirit of fair comparison, we compare them at an equal parameterization (size). Our scaling experiments are reserved for our $SBase$ models, which is intended to only be compared with subword T5 models, and not to unscaled byte-level baselines. Finally, we include an $SBase$ scaled version of Byte-level T5 for comparison.

**Setup**  We evaluate Base and SBase configurations of CHARFORMER with 203M and 134M parameters respectively. We compare to Base configurations of BERT and T5 that have a similar number of parameters. We pre-train all models on the C4 corpus for 1M steps using a batch size of $64$ and sequence length of $1024$. All non-subword models use a vocabulary of 256 bytes.[5] Our pre-training scheme corrupts spans with a mean length of 20 bytes. Each model is pre-trained on 16 TPU V3 chips. We pre-train our models with the Adafactor optimizer with an inverse square root learning rate. We then fine-tune on each individual task separately using a constant learning rate of $10^{-3}$. More details can be found in the Appendix.

---

[4]Compared to CANINE, Byte-level T5+LASC does not operate on Unicode codepoints and has a decoder. It thus forgoes character hash embeddings and upsampling procedures respectively.

[5]Following Xue et al. (2021) we discard illegal UTF-8 sequences and reuse the final 100 byte IDs as sentinel tokens.

Table 1: Comparison of CHARFORMER against other subword and character-level models with different parameter sizes on diverse standard English datasets.

| Model | $\|\theta\|$ | SST-2 | MNLI | QNLI | MRPC | QQP | STSB | COLA | AVG |
|---|---|---|---|---|---|---|---|---|---|
| BERT$_{Base,Subword}$ | 110M | 92.7 | 84.4/- | 88.4 | 86.7/- | - | - | - | - |
| T5$_{Base,Subword}$ | 220M | 92.7 | 84.2/84.6 | 90.5 | 88.9/92.1 | 91.6/88.7 | 88.0 | 53.8 | 84.3 |
| Byte-level T5$_{Base}$ | 200M | 91.6 | 82.5/82.7 | 88.7 | 87.3/91.0 | 90.9/87.7 | 84.3 | 45.1 | 81.5 |
| Byte-level T5+Conv$_{Base}$ | 205M | 89.8 | 81.1/82.5 | 89.2 | 83.6/89.2 | 90.7/87.7 | 85.0 | 47.1 | 81.2 |
| Byte-level T5+LASC$_{Base}$ | 205M | 90.0 | 80.0/80.8 | 87.1 | 82.8/88.1 | 89.0/85.4 | 83.7 | 25.3 | 77.0 |
| CHARFORMER$_{Base}$ | 203M | 91.6 | 82.6/82.7 | 89.0 | 87.3/91.1 | 91.2/88.1 | 85.3 | 42.6 | 81.4 |
| Byte-level T5$_{SBase}$ | 133M | 91.2 | 83.9/83.7 | 90.9 | 85.5/89.2 | 91.1/88.1 | 85.7 | 49.3 | 82.6 |
| CHARFORMER$_{SBase}$ | 134M | 91.5 | 83.7/84.4 | 91.0 | 87.5/91.4 | 91.4/88.5 | 87.3 | 51.8 | 83.6 |

Table 2: Results on comment classification on Civil Comments and Wiki Comments. Metrics are accuracy and AUC-PR. T5 baseline results are from (Tay et al., 2021).

| Model | Civil Comments | Wiki Comments |
|---|---|---|
| T5$_{Base,Subword}$ | 81.2 / - | 91.5 / - |
| Byte-level T5$_{Base}$ | 82.8 / 78.7 | 93.2 / 75.4 |
| Byte-level T5+LASC$_{Base}$ | 82.9 / 78.2 | 93.0 / 75.0 |
| CHARFORMER$_{Base}$ | 83.0 / 78.8 | 92.7 / 79.7 |
| CHARFORMER$_{SBase}$ | 83.0 / 78.9 | 93.5 / 75.5 |

Table 3: Results on text classification on long documents.

| Model | IMDb | News |
|---|---|---|
| T5$_{Base,Subword}$ | 94.2 | 93.5 |
| Byte-level T5$_{Base}$ | 91.5 | 93.6 |
| Byte-level T5+LASC$_{Base}$ | 91.1 | 93.5 |
| CHARFORMER$_{Base}$ | 91.5 | 94.0 |
| CHARFORMER$_{SBase}$ | 94.4 | 94.1 |

**Results** For all result tables, we divide the table into three sections: subword baseline(s), un-scaled byte-level baselines, and scaled CHARFORMER results. If a section and task combination has more than one model result, we underline the best result. We show result for GLUE in Table 1. CHARFORMER outperforms other character-level baselines trained under the same conditions with the same number of parameters across all tasks, while being considerably faster and requiring less compute than T5-style models that are directly applied to bytes or characters (see §4). CHARFORMER$_{SBase}$ performs even better despite having a smaller number of parameters compared to the Base configuration, demonstrating the usefulness of rescaling the transformer stack for character-level models. CHARFORMER$_{SBase}$ furthermore is the only model that performs on par or even outperforms the standard subword-based models on some tasks in standard English. In Table 3 we provide results for text classification of long documents. Here, CHARFORMER$_{SBase}$ is the only byte-level model to outperform T5$_{Base,Subword}$ on the IMDb classification task, and both CHARFORMER models outperform byte and subword level baselines on AGNews.

## 3.2 EXPERIMENTS ON NON-STANDARD ENGLISH DATASETS

The previous set of experiments demonstrated the ability of CHARFORMER to perform well on clean datasets consisting of standard English. However, character-level models are particularly suited to data that is noisy, containing spelling variations, typos, and other non-standard language.

**Data** To demonstrate CHARFORMER's ability to perform well on such data, we evaluate on toxicity detection using the Civil Comments (Borkan et al., 2019) and the Wikipedia Comments (Wulczyn et al., 2017) datasets. Both are standard benchmarks that require estimating the toxicity of user-generated content. We use the same setup as for the standard English datasets.

**Results** We show results in Table 2. Character-level models outperform the subword-based T5 model on both datasets, demonstrating their suitability to deal with such noisy, user-generated data. CHARFORMER achieves performs on par or outperforms other character-level methods on both datasets across the different model sizes.

## 3.3 MULTILINGUAL EXPERIMENTS

**Data** To evaluate the effectiveness of character-level models on multilingual data, we evaluate on standard cross-lingual question answering and classification tasks. In particular, we evaluate on the question answering tasks TyDiQA-GoldP (Clark et al., 2020), XQuAD (Artetxe et al., 2020), and MLQA (Lewis et al., 2020) as well as the natural language inference task XNLI (Conneau et al., 2018) and the paraphrase detection task PAWS-X (Yang et al., 2019) from XTREME (Hu et al., 2020). We evaluate on the in-language multi-task setting for TyDiQA-GoldP (Clark et al., 2020) where models

Table 4: Multilingual comparison of CHARFORMER against subword and byte-level models on in-language multi-task, translate-train multi-task, and cross-lingual zero-shot (training on English) settings. Model sizes are the same as those in Table 1. mBERT and mT5 baseline results are from (Xue et al., 2020).

| Model | $\|\theta\|$ | In-Language TyDiQA-GoldP | Translate-Train-All XQuAD | MLQA | XNLI | PAWS-X | Zero-Shot XNLI | PAWS-X |
|---|---|---|---|---|---|---|---|---|
| mBERT$_{Base}$ (Subword) | 179M | 77.6/68.0 | -/- | -/- | - | - | 65.4 | 81.9 |
| mT5$_{Base}$ (Subword) | 582M | 80.8/70.0 | 75.3/59.7 | 67.6/48.5 | 75.9 | 89.3 | 75.4 | 86.4 |
| Byte-level T5$_{Base}$ | 200M | 75.6/65.4 | 68.6/54.3 | 61.8/44.4 | 69.4 | 87.1 | 57.4 | 80.9 |
| Byte-level T5+LASC$_{Base}$ | 205M | 70.6/59.7 | 66.8/52.1 | 58.8/41.1 | 67.9 | 84.8 | 55.2 | 79.0 |
| CHARFORMER$_{Base}$ | 203M | 75.9/65.6 | 70.2/55.9 | 62.6/44.9 | 71.1 | 87.2 | 57.6 | 81.6 |
| CHARFORMER$_{SBase}$ | 134M | 79.1/68.8 | 73.6/59.0 | 66.3/48.5 | 72.2 | 88.2 | 66.6 | 85.2 |
| CHARFORMER$_{SBase,LongPT}$ | 134M | 81.2/71.3 | 74.2/59.8 | 67.2/49.4 | 72.8 | 88.6 | 67.8 | 83.7 |

Table 5: Comparison of pre-training compute metrics for mT5 (Subword) versus comparable quality CHARFORMER models on the mC4 dataset. 64 TPUv3 chips were used for this experiment. CHARFORMER$_{SBase}$ sees the same number of tokens after downsampling as mT5$_{Base}$, while CHARFORMER$_{SBase,LongPT}$ roughly sees the same amount of raw text as mT5$_{Base}$, given that a SentencePiece subword token is about 4.1 bytes on average (Xue et al., 2021). CHARFORMER$_{SBase}$ is 28% faster than mT5$_{Base}$, while using 33% of the FLOPS.

| Model | Batch Size | $L$ | $d_s$ | $\|\theta\|$ | Speed (steps/s) | FLOPS |
|---|---|---|---|---|---|---|
| mT5$_{Base}$ (Subword) | 1024 | 1024 | - | 582M | 1.54 | $1.3 \times 10^{15}$ |
| CHARFORMER$_{SBase}$ | 1024 | 2048 | 2 | 134M | 1.98 | $4.3 \times 10^{14}$ |
| CHARFORMER$_{SBase,LongPT}$ | 2048 | 2048 | 2 | 134M | 1.01 | $4.3 \times 10^{14}$ |

are fine-tuned on the combined gold data in all target languages and the translate-train-all setting where models are fine-tuned on English training data plus translations in all target languages for the other datasets. Both are the best-performing settings for the respective tasks in (Hu et al., 2020). In addition, we evaluate on zero-shot cross-lingual transfer from English on XNLI and PAWS-X.

**Baselines** We compare to strong multilingual subword-based baselines including multilingual BERT (Devlin et al., 2019) and multilingual T5 (Xue et al., 2020). In addition, we compare to the byte-level models from §3.1, which we pre-train on multilingual data.

**Setup** We pre-train CHARFORMER as well as the Byte-level T5 and Byte-level T5+LASC baselines on multilingual mC4 Common Crawl (Xue et al., 2020) in 101 languages. Base size models were trained for 1M steps using a batch size of 64 and sequence length of 2048, with the exception of Byte-level T5$_{Base}$, which was trained with a sequence length of 1024, as training speed was prohibitively slow (see Table 11). CHARFORMER$_{SBase}$ and CHARFORMER$_{SBase,LongPT}$ (longer pre-training) are trained with larger batch sizes for fair comparison with mT5. In particular, CHARFORMER$_{SBase}$ pre-trains on the same amount of tokens after downsampling as mT5$_{Base}$, while CHARFORMER$_{SBase,LongPT}$ pre-trains on roughly the same amount of raw text as mT5$_{Base}$, given that a SentencePiece subword token is about 4.1 bytes on average (Xue et al., 2021); see Table 5 for further details. All models were fine-tuned with an input sequence length of 4096 for question-answering tasks and 2048 for inference tasks. Score calibration was not used for these experiments, as it did not benefit the model in the multilingual setting. For XNLI and PAWS-X (both translate-train and zero-shot settings), we also observed that performance improved if the GBST layer was not updated during fine-tuning; the reported CHARFORMER numbers reflect this configuration. Otherwise, all other hyper-parameters and model sizes are unchanged from the English experimental setup.

**Results** We show in-language multi-task, translate-train, and cross-lingual zero-shot results in Table 4. CHARFORMER$_{SBase}$ is competitive with standard subword-based models and CHARFORMER$_{SBase,LongPT}$ outperforms subword-based models on TyDiQA-GoldP (in-language multi-task). Additionally, in the translate-train setting CHARFORMER$_{SBase,LongPT}$ is on par with subword models on XQuAD and MLQA, and close to parity on PAWS-X. Furthermore, CHARFORMER outperforms other character-level models in the zero-shot setting. However, we observe that this setting still remains a challenge for token-free models in general. We hypothesize that model size may be a major factor here. Finally, we provide additional comparison between GBST and LASC at a fixed down-sampling rate in Section 8.4 (Appendix), showing that GBST significantly outperforms LASC on TyDiQA.

Table 6: Pre-training compute metrics of models at different input lengths, downsampling rates, and model sizes on the English C4 dataset. 16 TPUv3 chips were used for this experiment. These numbers reflect a batch size of 64. Memory refers to per-device peak memory usage on TPUv3 chips.

| Model | $L$ | $d_s$ | $|\theta|$ | Speed (steps/s) | FLOPS | Peak Mem. |
|---|---|---|---|---|---|---|
| T5$_{Base}$ (Subword) | 512 | - | 220M | 9.3 | $1.1 \times 10^{13}$ | - |
| Byte-level T5$_{Base}$ | 1024 | 1 | 200M | 8.2 | $2.9 \times 10^{13}$ | 3.09GB |
| Byte-level T5+LASC$_{Base}$ | 1024 | 4 | 205M | 15 | $9.9 \times 10^{12}$ | 1.62GB |
| CHARFORMER$_{Base}$ | 1024 | 2 | 206M | 11 | $1.6 \times 10^{13}$ | 1.95GB |
| CHARFORMER$_{Base}$ | 1024 | 3 | 203M | 15 | $1.1 \times 10^{13}$ | 1.63GB |
| CHARFORMER$_{SBase}$ | 1024 | 2 | 134M | 14 | $1.3 \times 10^{13}$ | 1.73GB |
| CHARFORMER$_{SBase}$ | 1024 | 3 | 134M | 20 | $8.7 \times 10^{12}$ | 1.34GB |

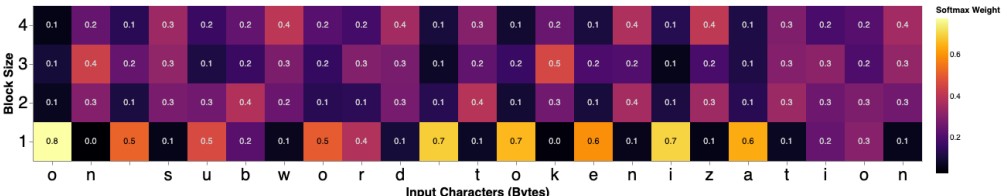

Figure 2: Visualization of block scores (softmax weights) for every byte position from multilingual CHARFORMER$_{SBase}$ on an example English input.

## 4 SPEED, MEMORY AND PARAMETERS

Table 6 reports the speed (global training steps per second), parameter sizes and number of floating point operations (FLOPS) for each forward pass of the models used in our experiments. All experiments were run on 16 TPU-v3 chips and speed is benchmarked on English C4 pre-training at the 1K input length ($L$). CHARFORMER models are generally more efficient both in terms of speed and FLOPS compared to other character-level models at different parameter sizes. With a low down-sampling rate $d_s$ for CHARFORMER, Byte-level T5+LASC is more efficient due to using a higher down-sampling rate. Directly consuming the character sequence with a Transformer model is slow and requires a large number of FLOPS, which is exacerbated with longer sequence lengths where Byte-level T5 is more than $2\times$ slower than the fastest CHARFORMER. This difference is even larger at longer input sequence lengths, which we report in the Appendix. CHARFORMER$_{SBase}$ achieves better performance (see §3) with fewer parameters but more FLOPS by using a deep thin encoder and is twice as fast as the subword-based model with similar performance, T5$_{Base}$.

## 5 VISUALIZING LATENT SUBWORDS

One benefit of CHARFORMER compared to other character-level methods is that the subwords it learns are directly interpretable and may give some indications to the behaviour of the underlying model. We visualize the scores the multilingual CHARFORMER has learned to assign to subword blocks of different sizes for the string 'on subword tokenization' in Figure 2. We observe that the model learns to allocate single-character subword blocks predominantly to vowels and whitespace in English. Moreover, in English the model allocates larger subword blocks to the beginning and end consonants of a subword. Together, we believe this suggests that the model has learned a meaningful segmentation of the input, and that it is able to dynamically mix between byte-level and subword-level features. Such behaviour could also parallel the relative importance attributed to consonants for word identification observed during reading in humans (Lee et al., 2001; Carreiras et al., 2008).

## 6 RELATED WORK

**Subword tokenization** Standard algorithms for *deterministic* subword tokenization are Byte Pair Encoding (BPE; Sennrich et al., 2016), Wordpiece (Wu et al., 2016), and SentencePiece (Kudo and Richardson, 2018). Prior work has highlighted issues with some of these algorithms (Bostrom and Durrett, 2020) and has generally observed that models learned with such rigid tokenization do not cope well with variation in language (Sun et al., 2020). To make a model more robust to morphological and compositional generalization, *probabilistic* segmentation algorithms such as

subword regularization (Kudo, 2018) and BPE-dropout (Provilkov et al., 2020) have been proposed, which sample different segmentations during training. Recent methods propose to make models more robust for downstream tasks by enforcing prediction consistency between deterministic and probabilistic segmentations (Wang et al., 2021) and propose to update the tokenizer based on the downstream loss under different segmentations (Hiraoka et al., 2020; 2021). He et al. (2020) proposed DPE (dynamic programming encoding), a segmentation-based tokenization algorithm based on dynamic programming. Such methods, however, incur large computation costs due multiple forward passes needing to be performed for each segmentation of an example or due to the expensive DP computation, which make them unsuitable for pre-training.

**Character-level models**    For recurrent neural networks, pure character-level models that take a sequence of characters as input (Graves, 2013; Zhang et al., 2015; Hwang and Sung, 2017) have mostly been superseded by *character-aware* methods that compute a token-level representation using a CNN over characters (Kim et al., 2016; Jozefowicz et al., 2016; Peters et al., 2018) due to poor performance when learning directly from characters. Such character-aware representations have lately been applied to deep Transformer models (El Boukkouri et al., 2020; Ma et al., 2020). These methods, however, still require tokenization for pre-processing and cannot be directly applied to languages without whitespace separation. Prior work also learned segmentation as part of the model but did not scale very well (Wang et al., 2017; Kreutzer and Sokolov, 2018; Kawakami et al., 2019). One notable exception is (Lee et al., 2017), which enabled fully character-level neural machine translation, using stacked convolutions, max pooling, and highway networks. Building on this, recent *tokenization-free* approaches such as CANINE (Clark et al., 2021) revisit the original character-level setting in the context of large pre-trained language models with a focus on multilingual models. Our method outperforms CANINE-style downsampling (local attention, strided convolutions) and also leads to improvements in the monolingual setting, while using less compute and parameters to down-sample than both Lee et al. (2017) and Clark et al. (2021). Recently, ByT5 (Xue et al., 2021) set new start-of-the-art results for tokenization-free models, by operating on the byte-level. This work performs on par with or outperforms ByT5, with significant gains in speed and compute efficiency.

**Multilingual models**    Current multilingual models are generally analogues to successful monolingual Transformer models (Ruder et al., 2021). Consequently, models such as multilingual BERT (Devlin et al., 2019) and XLM-R (Conneau et al., 2020) employ the same subword tokenization algorithms as monolingual models, now applied to a massively multilingual corpus. In the multilingual setting, the problems of subword-based tokenization are exacerbated as tokens in languages with few data are over-segmented while high-frequency tokens are under-segmented, which limits cross-lingual transfer (Wang et al., 2021). This motivates our work as well as recent work on character-level models.

**Efficient Transformers**    Moving from subwords to characters significantly increases the sequence length, which is an issue for Transformers due to the quadratic complexity of self-attention. Many efficient self-attention models have been proposed (Choromanski et al., 2020; Wang et al., 2020; Zaheer et al., 2020) to tackle this problem; see (Tay et al., 2020b;a) for a comprehensive overview. Notably, the CANINE model uses local attention (Parmar et al., 2018), which could also be swapped with another efficient Transformer variant. We note that the problem of efficiency is important but not the only challenge towards developing performant tokenization-free models. While applying an efficient attention mechanism might solve the fundamental computational costs of employing character-level models, there is no guarantee that these models will learn locally meaningful compositions.

## 7    CONCLUSION

We have proposed CHARFORMER, a re-scaled Transformer architecture that integrates gradient-based subword tokenization, a novel lightweight tokenization method that enables efficient end-to-end learning of latent subwords directly from characters. We have demonstrated that English and multilingual variants of CHARFORMER outperform strong character-level baselines across various datasets while being more efficient. CHARFORMER achieves performance on par with subword-based models on standard English tasks and outperforms subword-based models on noisy social media data. On multilingual data, CHARFORMER generally performs on par with subword-based models, while being faster than both byte-level and subword-level baselines. Finally, we provide a method to inspect the inner workings of the GBST module. Overall, we believe that the strong results presented in this paper pave the way for highly effective and powerful token-free models.

## ACKNOWLEDGEMENTS

We would like to thank Jon Clark, Noah Constant, and Kris Cao for valuable feedback on drafts of this manuscript.

## ETHICS STATEMENT

Standard subword tokenization algorithms produce segmentations that do not equally represents words and phrases in different languages. Instead, they are biased towards languages that already have many resources available, which leads to multilingual models performing worse on under-represented languages (Wang et al., 2021). Tokenization-free approaches such as the one proposed in this paper may help to ameliorate this to some extent. Another challenge to using large multilingual models in practice is their relative computational inefficiency, which makes them unsuitable in resource-constrained settings common in scenarios where under-represented languages are spoken. CHARFORMER trains 28% faster than mT5 and has $3\times$ fewer parameters, so may be a more suitable choice in such settings compared to state-of-the-art multilingual models.

## REPRODUCIBILITY STATEMENT

All code to train the core byte-level Transformer encoder-decoder for CHARFORMER its variants is already open-sourced as a part of the Mesh Tensorflow[6] (Shazeer et al., 2018), T5[7] (Raffel et al., 2020), and ByT5[8] (Xue et al., 2021) libraries. Additionally, an implementation of Charformer GBST compatible with existing open-source models has been open-sourced[9]. We also include a simplified Tensorflow implementation of GBST in Section 8.7 of the Appendix. All detailed experiment and hyperparameter settings required to reproduce our experiments can be found in Section 8.2 of the Appendix.

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

# 8 APPENDIX

## 8.1 OVERVIEW

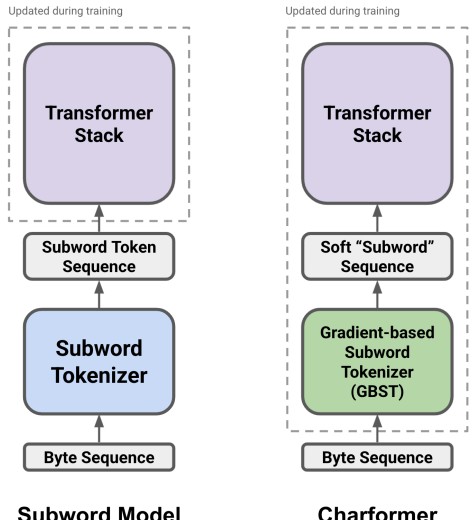

Figure 3: High-level differences between traditional subword Transformer models and Charformer which uses gradient-based subword tokenization.

## 8.2 HYPERPARAMETERS

This section describes the hyperparameters that we use in our experiments.

**Monolingual English Datasets** Our small model follows the T5 small model size with 6 encoder layers and 6 decoder layers, hidden size $d_{model}$ of 512, 8 heads, $d_{kv}$ of 32 and $d_{ff}$ of 2048. This corresponds to *bi_v1_small.gin* in the T5 codebase. The base model (corresponding to *bi_v1.gin*) has 12 encoder layers, 12 decoder layers, $d_{model}$ of 768, $d_{ff}$ of 3072 and 12 heads. The SBase model has 24 encoder layers and 6 decoder layers, while the remainder of its hyperparameters remain identical to the small model. All Transformer stacks use relative attention over positional encodings as per (Raffel et al., 2020). For pre-training, we run our models for $1M$ steps on C4 with a batch size of 64. The maximum sequence length for all tasks is set to $1024$. TPU packing is not activated for Charformer. For Charformer, the filter size of the pre-GBST convolution is set to 5 by default. For CHARFORMER, the downsampling rate is tuned in the range of $\{2, 3, 4\}$. For smaller models, the rate of 2 seems to work consistently the best. For base models, the best models used a downsampling rate of either 2 or 3. For the SBase models, the optimal downsampling rate was often 3.

**Multilingual Datasets** Hyperparameters are kept constant between English and multilingual tasks except for the following differences. For pre-training, we run our models for 1M steps with a batch size of 64, except for CHARFORMER$_{SBase}$ which uses a batch size of 1024 and CHAR-FORMER$_{SBase,LongPT}$ which usees a batch size of 2048. Models were pre-trained with a maximum sequence length of 2048 and fine-tuned with a maximum sequence length of 4096 for TyDiQA, XQuAD, and MLQA, and 2048 for XNLI and PAWS-X. Byte-level T5$_{Base}$ was the only model to be pre-trained with a maximum sequence length of 1024, as it was prohibitively slow, see Table 11. Fine-tuning and inference for this model, however still used 4096 and 2048 input lengths identical to other models. For all tasks, CHARFORMER models used a downsampling rate of 2, while Byte-level T5+LASC models used a downsampling rate of 4 (Clark et al., 2021). The downsampling rate of 2 was picked by ablating the downsampling rate on the TyDiQA-GoldP validation set. CHARFORMER models for XNLI and PAWS-X additionally did not back-propagate into the GBST layer during fine-tuning. Checkpoints were picked based on the dev set metrics, and then evaluated on test set. Reported metrics represent the macro-average of all languages in the task.

## 8.3 ABLATION STUDY

This section presents our ablation experiments for both English and multilingual tasks. We analyze the impact of various hyper-parameters and modeling choices such as using offsets vs 1D convolutions. Across experiments, we find that pre-GBST convolutions are preferred to enumerating offset blocks, as it results in similar (or better) quality but a more efficient implementation. For English tasks, block score calibration (BC) improves performance. We note that in the multilingual setting, block score calibration has little effect. The impact of different downsampling rates varies across tasks and model sizes. We also experimented with different convolution filter sizes in English and found that they did not significantly impact performance. Likewise, using a different character span corruption rate during pre-training did not significantly impact performance. Adding feed-forward layers to the CHARFORMER module in similar fashion to a Transformer block was also not obviously helpful.

Table 7: Ablation studies with CHARFORMER$_{Small}$ on English tasks.

| Ablation | $d_s$ | Size | SST-2 | MNLI$_{mm}$ | IMDb |
|---|---|---|---|---|---|
| Offsets | 2 | S | 89.11 | 79.50 | 90.49 |
| Conv | 2 | S | 89.11 | 79.65 | **90.63** |
| Conv + BC | 2 | S | 89.56 | **80.15** | 90.60 |
| Conv + Offsets + BC | 2 | S | 89.11 | 79.68 | 90.48 |
| Conv | 3 | S | 89.45 | 80.07 | 90.15 |
| Conv | 4 | S | 89.11 | 79.82 | 90.21 |
| Conv | 2 | B | 90.60 | 82.92 | 91.46 |
| Conv | 3 | B | 91.40 | 82.74 | 91.46 |
| Conv | 4 | B | 91.40 | 82.67 | 92.33 |

Table 8: Effect of freezing the GBST layer for XNLI and PAWS-X.

| Model | $d_s$ | Freeze GBST | XNLI (Zero) | XNLI (Translate) | PAWS-X (Zero) | PAWS-X (Translate) |
|---|---|---|---|---|---|---|
| CHARFORMER$_{Small}$ | 2 | No | 44.5 | 62.7 | 27.9 | 37.5 |
| CHARFORMER$_{Small}$ | 2 | Yes | 50.9 | 68.7 | 77.1 | 84.8 |
| CHARFORMER$_{Small}$ | 3 | No | 47.9 | 67.9 | 29.5 | 36.8 |
| CHARFORMER$_{Small}$ | 3 | Yes | 43.2 | 68.6 | 77.8 | 83.7 |
| CHARFORMER$_{Small}$ | 4 | No | 47.5 | 47.5 | 30.9 | 36.9 |
| CHARFORMER$_{Small}$ | 4 | Yes | 43.6 | 43.6 | 77.9 | 83.5 |

## 8.4 COMPARING DOWNSAMPLING APPROACHES

In Table 10, we compare GBST downsampling with LASC downsampling (Clark et al., 2021) on TyDiQA-GoldP. For this experiment we use the same hyperparameters as in Section 3.3, except the pre-training input length is 1024 instead of 2048. Note that this difference is negligible (0.1 F1) for CHARFORMER$_{Base}$, $d_s = 2$ which also appears in Table 4. All hyperparameters are fixed between CHARFORMER and Byte-level T5+LASC. Following (Clark et al., 2021) we set $d_s = 4$ for LASC, and we compare CHARFORMER at the same downsampling rate. We additionally include $d_s = 2$ and $d_s = 3$ for CHARFORMER for comparison. With the same hyperparameters and downsampling rate, CHARFORMER outperforms Byte-level T5+LASC on TyDiQA-GoldP.

Table 9: Effect of $d_s$ on TyDiQA-GoldP (in-language multi-task).

| Model | $d_s$ | TyDiQA-GoldP F1 |
|---|---|---|
| CHARFORMER$_{Small}$ | 2 | 69.6 |
| CHARFORMER$_{Small}$ | 3 | 68.1 |
| CHARFORMER$_{Small}$ | 4 | 66.6 |
| Byte-level T5+LASC$_{Small}$ | 4 | 64.9 |
| CHARFORMER$_{Base}$ | 2 | 75.8 |
| CHARFORMER$_{Base}$ | 3 | 74.3 |
| CHARFORMER$_{Base}$ | 4 | 73.2 |
| Byte-level T5+LASC$_{Base}$ | 4 | 70.6 |

## 8.5 LARGE-SCALE EXPERIMENTS

In this section we report preliminary results for scaling Charformer using the same number of parameters as mT5$_{Large}$ and ByT5$_{Large}$ (1.23B). We follow a model scaling configuration identical to ByT5 in these experiments, and use the same hyperparameter settings as our main multilingual results.

Table 10: Comparison on TyDiQA at 1.23B parameters. *Due to resource constraints, the Charformer result below uses ~100K less pretraining steps than ByT5 and mT5.

| Model | TyDiQA-GoldP F1 / EM |
|---|---|
| mT5$_{Large}$ | 85.3 / 75.3 |
| ByT5$_{Large}$ | 87.7 / 79.2 |
| CHARFORMER* | 86.3 / 77.3 |

**Results** The CHARFORMER model under the same scaling as ByT5$_{Large}$ was able to outperform mT5$_{Large}$, a very strong baseline. Our preliminary results at this scale shows that CHARFORMER is competitive with, but is 1.4 F1 behind ByT5$_{Large}$. However, we point out two important notes. First, the CHARFORMER result is undertrained compared to ByT5$_{Large}$ since 10% of the pretraining has not finished. Second, the CHARFORMER model is also twice as fast as ByT5, as seen from Table 11.

## 8.6 MULTILINGUAL EXPERIMENTS

This section contains detailed results for our multilingual experiments.

Table 11: Compute metrics of base models at longer (2K) input length on the mC4 pre-training corpus, using a batch size of 64 on 16 TPU-v3 chips.

| Model | $L$ | $d_s$ | $|\theta|$ | Speed (steps/s) | FLOPS |
|---|---|---|---|---|---|
| Byte-level T5$_{Base}$ | 2048 | 1 | 200M | 2.7 | $2.0 \times 10^{13}$ |
| Byte-level T5+LASC$_{Base}$ | 2048 | 4 | 205M | 11 | $5.5 \times 10^{12}$ |
| CHARFORMER$_{Base}$ | 2048 | 2 | 203M | 6.1 | $9.5 \times 10^{12}$ |
| CHARFORMER$_{Base}$ | 2048 | 3 | 203M | 10 | $6.5 \times 10^{12}$ |
| CHARFORMER$_{SBase}$ | 2048 | 2 | 134M | 6.1 | $9.2 \times 10^{12}$ |

Table 12: Per-language breakdown of in-language multi-task TyDiQA-GoldP results.

| Model | $|\theta|$ | ar | bn | en | fi | id | ko | ru | sw | te | Avg. |
|---|---|---|---|---|---|---|---|---|---|---|---|
| mBERT$_{Base}$ (Subword) | 179M | -/- | -/- | -/- | -/- | -/- | -/- | -/- | -/- | -/- | 77.6/68.0 |
| mT5$_{Base}$ (Subword) | 582M | 84.2/71.8 | 80.0/69.0 | 76.6/65.2 | 80.1/69.3 | 85.5/75.0 | 70.3/61.6 | 77.5/64.4 | 83.6/74.9 | 88.2/78.0 | 80.8 / 70.0 |
| Byte-level T5$_{Base}$ | 200M | 81.4/67.0 | 66.8/56.6 | 69.8/59.5 | 75.6/63.0 | 81.6/72.4 | 64.6/58.7 | 74.1/60.8 | 81.8/74.3 | 85.0/76.1 | 75.6/65.4 |
| Byte-level T5+LASC$_{Base}$ | 205M | 78.1/62.3 | 61.1/50.4 | 66.7/55.2 | 72.5/60.4 | 79.9/68.3 | 51.5/43.5 | 70.4/58.7 | 74.7/67.5 | 80.2/71.2 | 70.6/59.7 |
| CHARFORMER$_{Base}$ | 203M | 81.8/67.9 | 69.1/60.2 | 71.4/60.5 | 76.3/64.2 | 83.0/73.1 | 62.7/54.3 | 74.7/61.7 | 80.2/73.3 | 83.6/75.0 | 75.9/65.6 |
| CHARFORMER$_{SBase}$ | 134M | 82.4/68.1 | 78.1/67.3 | 75.4/64.3 | 79.5/68.2 | 85.0/75.9 | 66.6/58.0 | 77.0/64.3 | 81.5/74.1 | 86.5/78.6 | 79.1/68.8 |
| CHARFORMER$_{SBase,LongPT}$ | 134M | 85.7/74.5 | 78.7/67.3 | 76.8/65.9 | 81.9/70.6 | 86.7/79.1 | 69.4/61.6 | 79.2/67.1 | 83.7/75.2 | 88.8/80.6 | 81.2/71.3 |

Table 13: Per-language breakdown of translate-train-all XQuAD results.

| Model | $|\theta|$ | ar | de | el | en | es | hi | ru | th | tr | vi | zh | Avg. |
|---|---|---|---|---|---|---|---|---|---|---|---|---|---|
| mT5$_{Base}$ (Subword) | 582M | 72.4/55.2 | 76.9/59.7 | 76.8/58.8 | 83.1/70.3 | 79.0/61.2 | 71.4/53.4 | 76.1/58.5 | 67.9/62.0 | 72.5/51.4 | 75.9/56.3 | 76.9/69.7 | 75.3/59.7 |
| Byte-level T5$_{Base}$ | 200M | 64.8/47.9 | 74.3/58.3 | 69.2/51.8 | 81.5/70.4 | 77.2/60.4 | 67.0/51.5 | 72.3/55.5 | 48.3/41.9 | 69.6/51.7 | 73.3/54.4 | 57.3/53.3 | 68.6/54.3 |
| Byte-level T5+LASC$_{Base}$ | 205M | 62.9/45.5 | 70.6/54.2 | 68.3/52.3 | 80.1/68.4 | 74.8/57.9 | 63.1/46.2 | 68.2/52.2 | 50.0/43.4 | 67.1/48.2 | 71.7/51.8 | 57.7/52.7 | 66.8/52.1 |
| CHARFORMER$_{Base}$ | 203M | 65.7/49.8 | 74.2/58.0 | 71.1/53.1 | 82.2/70.5 | 77.8/61.0 | 67.0/51.3 | 73.4/57.6 | 54.3/48.0 | 70.3/53.0 | 74.6/55.6 | 62.0/56.6 | 70.2/55.9 |
| CHARFORMER$_{SBase}$ | 134M | 70.3/53.7 | 78.6/61.4 | 74.4/55.1 | 85.1/73.7 | 79.8/63.6 | 69.1/52.7 | 76.7/61.3 | 57.6/51.2 | 73.9/55.8 | 76.8/57.6 | 67.4/62.4 | 73.6/59.0 |
| CHARFORMER$_{SBase,LongPT}$ | 134M | 72.6/55.0 | 79.0/62.3 | 74.9/56.1 | 85.4/74.5 | 80.4/63.4 | 70.6/56.1 | 77.8/62.2 | 56.1/49.2 | 76.1/58.2 | 77.7/59.4 | 66.0/61.8 | 74.2/59.8 |

Table 14: Per-language breakdown of translate-train-all MLQA results.

| Model | $|\theta|$ | ar | de | en | es | hi | vi | zh | Avg. |
|---|---|---|---|---|---|---|---|---|---|
| mT5$_{Base}$ (Subword) | 582M | 61.1/40.7 | 65.5/49.2 | 80.7/66.3 | 70.7/52.1 | 63.6/44.3 | 68.0/47.6 | 63.5/39.4 | 67.6/48.5 |
| Byte-level T5$_{Base}$ | 200M | 52.6/34.2 | 60.5/46.1 | 77.7/64.8 | 67.1/49.2 | 52.9/36.5 | 63.6/43.8 | 58.3/36.4 | 61.8/44.4 |
| Byte-level T5+LASC$_{Base}$ | 205M | 50.8/32.0 | 58.1/43.5 | 75.8/62.2 | 64.7/46.7 | 49.2/32.6 | 60.4/40.4 | 52.6/30.6 | 58.8/41.1 |
| CHARFORMER$_{Base}$ | 203M | 53.5/34.5 | 61.3/46.8 | 78.5/65.4 | 67.2/49.3 | 54.5/37.6 | 64.3/43.9 | 58.8/36.6 | 62.6/44.9 |
| CHARFORMER$_{SBase}$ | 134M | 58.3/39.1 | 65.7/50.5 | 81.8/68.7 | 71.0/53.1 | 57.7/40.8 | 67.3/46.8 | 62.7/40.8 | 66.3/48.5 |
| CHARFORMER$_{SBase,LongPT}$ | 134M | 59.6/40.0 | 66.6/51.3 | 82.2/69.0 | 72.1/54.5 | 59.7/42.9 | 68.2/47.4 | 62.4/40.7 | 67.2/49.4 |

Table 15: Per-language breakdown of translate-train-all and cross-lingual zero-shot XNLI results.

| Model | $|\theta|$ | ar | bg | de | el | en | es | fr | hi | ru | sw | th | tr | ur | vi | zh | Avg. |
|---|---|---|---|---|---|---|---|---|---|---|---|---|---|---|---|---|---|
| | | | | | | *Translate-Train-All* | | | | | | | | | | | |
| mT5$_{Base}$ (Subword) | 582M | 74.4 | 78.5 | 77.7 | 78.1 | 82.0 | 79.1 | 77.9 | 72.2 | 76.5 | 71.5 | 75.0 | 74.8 | 70.4 | 74.5 | 76.0 | 75.9 |
| Byte-level T5$_{Base}$ | 200M | 67.1 | 72.0 | 71.0 | 70.6 | 76.9 | 74.0 | 73.4 | 63.7 | 69.2 | 66.2 | 65.7 | 69.4 | 62.8 | 69.6 | 69.0 | 69.4 |
| Byte-level T5+LASC$_{Base}$ | 205M | 65.6 | 72.1 | 70.5 | 67.9 | 75.6 | 73.4 | 72.2 | 63.5 | 68.6 | 65.4 | 64.5 | 67.4 | 62.4 | 68.3 | 61.0 | 67.9 |
| CHARFORMER$_{Base}$ | 203M | 69.5 | 72.9 | 72.7 | 72.6 | 78.2 | 74.5 | 73.6 | 67.0 | 71.7 | 67.9 | 68.1 | 70.8 | 65.0 | 70.7 | 71.5 | 71.1 |
| CHARFORMER$_{SBase}$ | 134M | 70.8 | 75.7 | 75.9 | 73.1 | 80.9 | 76.9 | 76.8 | 65.6 | 74.7 | 65.7 | 67.7 | 72.0 | 63.1 | 72.9 | 71.5 | 72.2 |
| CHARFORMER$_{SBase,LongPT}$ | 134M | 71.1 | 75.9 | 73.6 | 74.2 | 80.8 | 76.6 | 76.8 | 69.2 | 72.2 | 68.2 | 71.0 | 71.2 | 65.7 | 72.9 | 73.0 | 72.8 |
| | | | | | | *Cross-Lingual Zero-Shot* | | | | | | | | | | | |
| mBERT$_{Base}$ (Subword) | 179M | 64.3 | 68.0 | 70.0 | 65.3 | 80.8 | 73.5 | 73.4 | 58.9 | 67.8 | 49.7 | 54.1 | 60.9 | 57.2 | 69.3 | 67.8 | 65.4 |
| mT5$_{Base}$ (Subword) | 582M | 73.3 | 78.6 | 77.4 | 77.1 | 84.7 | 80.3 | 79.1 | 70.8 | 77.1 | 69.4 | 73.2 | 72.8 | 68.3 | 74.2 | 74.1 | 75.4 |
| Byte-level T5$_{Base}$ | 200M | 56.7 | 61.2 | 63.0 | 60.9 | 79.2 | 70.1 | 65.3 | 43.9 | 61.0 | 45.5 | 43.5 | 52.0 | 44.3 | 58.3 | 55.6 | 57.4 |
| Byte-level T5+LASC$_{Base}$ | 205M | 53.3 | 58.8 | 62.2 | 54.9 | 77.1 | 68.6 | 65.4 | 44.7 | 58.4 | 46.1 | 43.6 | 50.4 | 42.8 | 55.9 | 46.1 | 55.2 |
| CHARFORMER$_{Base}$ | 203M | 55.7 | 61.1 | 64.8 | 60.1 | 77.3 | 69.9 | 67.9 | 44.4 | 60.2 | 45.3 | 47.9 | 54.0 | 43.5 | 59.1 | 53.4 | 57.6 |
| CHARFORMER$_{SBase}$ | 134M | 66.4 | 71.0 | 72.7 | 68.6 | 82.4 | 77.1 | 75.4 | 57.6 | 70.6 | 48.7 | 61.4 | 61.8 | 54.1 | 68.9 | 62.8 | 66.6 |
| CHARFORMER$_{SBase,LongPT}$ | 134M | 68.4 | 70.9 | 74.3 | 70.2 | 82.4 | 77.0 | 76.6 | 59.9 | 71.0 | 42.6 | 64.0 | 65.5 | 56.5 | 71.2 | 66.0 | 67.8 |

Table 16: Per-language breakdown of translate-train-all and cross-lingual zero-shot PAWS-X results.

| Model | $|\theta|$ | de | en | es | fr | ja | ko | zh | Avg. |
|---|---|---|---|---|---|---|---|---|---|
| | | | | *Translate-Train-All* | | | | | |
| mT5$_{Base}$ (Subword) | 582M | 90.9 | 95.5 | 91.4 | 92.5 | 83.6 | 84.8 | 86.4 | 89.3 |
| Byte-level T5$_{Base}$ | 200M | 89.3 | 94.6 | 90.1 | 90.3 | 81.4 | 81.1 | 82.3 | 87.0 |
| Byte-level T5+LASC$_{Base}$ | 205M | 87.3 | 93.1 | 89.2 | 89.2 | 81.0 | 72.9 | 80.8 | 84.8 |
| CHARFORMER$_{Base}$ | 203M | 89.9 | 94.6 | 89.8 | 91.4 | 82.7 | 78.4 | 83.3 | 87.2 |
| CHARFORMER$_{SBase}$ | 134M | 89.9 | 95.9 | 91.8 | 92.2 | 83.9 | 78.9 | 84.4 | 88.2 |
| CHARFORMER$_{SBase,LongPT}$ | 134M | 90.7 | 95.1 | 92.2 | 92.2 | 84.1 | 81.6 | 84.6 | 88.6 |
| | | | | *Cross-Lingual Zero-Shot* | | | | | |
| mBERT$_{Base}$ (Subword) | 179M | 85.7 | 94.0 | 87.4 | 87.0 | 73.0 | 69.6 | 77.0 | 81.9 |
| mT5$_{Base}$ (Subword) | 582M | 89.4 | 95.4 | 89.6 | 91.2 | 79.8 | 78.5 | 81.1 | 86.4 |
| Byte-level T5$_{Base}$ | 200M | 84.7 | 93.8 | 85.8 | 86.4 | 72.2 | 67.9 | 75.2 | 80.9 |
| Byte-level T5+LASC$_{Base}$ | 205M | 83.2 | 93.2 | 84.1 | 85.0 | 67.9 | 66.4 | 73.4 | 79.0 |
| CHARFORMER$_{Base}$ | 203M | 86.1 | 94.8 | 87.2 | 88.0 | 70.1 | 69.7 | 75.5 | 81.6 |
| CHARFORMER$_{SBase}$ | 134M | 89.6 | 95.2 | 90.7 | 90.7 | 77.1 | 74.4 | 78.9 | 85.2 |
| CHARFORMER$_{SBase,LongPT}$ | 134M | 89.8 | 95.3 | 88.7 | 89.7 | 74.5 | 68.9 | 78.9 | 83.7 |

## 8.7 EXAMPLE IMPLEMENTATION

For additional clarity, we include a simplified implementation of the GBST module in Tensorflow below. Default hyper-parameters here match those used in the paper.

```python
from typing import Optional

import tensorflow as tf

keras_layers = tf.keras.layers

class GBSTLayer(keras_layers.Layer):
  """Performs Charformer GBST on a sequence.

  Attributes:
    input_shape: Shape [len, embedding_size] of input tensor in future calls,
      without batch dimension.
    downsample_rate: Integer of how much to downsample by.
```

```python
    max_subword_block_width: Integer of max block size to use for enumeration.
    block_attention: Hhether to use block score calibration.
    block_scoring_network: module for parameterized block scoring.
    conv_kernel_size: Integer of the size of the pre-GBST convolution kernel.
  """

  def __init__(self,
               input_shape: tf.Tensor,
               downsample_rate: int = 2,
               max_subword_block_width: int = 4,
               block_attention: bool = False,
               conv_kernel_size: Optional[int] = 5):
    super(GBSTLayer, self).__init__()
    self.downsample_rate = downsample_rate
    self.max_subword_block_width = max_subword_block_width
    self.conv_kernel_size = conv_kernel_size
    self.conv_layer = keras_layers.Conv1D(
        input_shape[-1], self.conv_kernel_size, input_shape=input_shape)
    self.block_attention = block_attention
    self.block_scoring_network = keras_layers.Dense(1, use_bias=False)

  def call(self, inputs):
    """Performs downsampling on the character-scale input representation.

    Args:
      inputs: float Tensor of shape [batch_size, seq_length,
        embedding_size].

    Returns:
      <float>[batch_size, seq_length / downsample_rate , embedding_size].
        Downsampled sequences.
    """
    length = inputs.shape[1]

    if self.conv_kernel_size:
      inputs = self.conv_layer(inputs)

    all_block_scores = []
    all_sequences = []
    for subword_len in range(1, self.max_subword_block_width):
      padded_input = inputs
      # Pad the sequence length if needed.
      if length % subword_len != 0:
        pad_amt = subword_len - int(length % subword_len)
        padding = tf.constant([[0, 0], [0, pad_amt], [0, 0]])
        padded_input = tf.pad(inputs, padding)

      # For this block size, form candidate block embeddings and scores.
      # candidates shape: [batch, seq_len/subword_len, dim]
      # block_scores shape: [batch, seq_len/subword_len, 1]
      candidates = tf.nn.avg_pool(
          padded_input, [subword_len], strides=[subword_len], padding="VALID")
      block_scores = self.block_scoring_network(candidates)

      # Upsample it back to the original sequence length.
      retiled_seq = tf.repeat(candidates, subword_len, axis=1)
      retiled_block_scores = tf.repeat(block_scores, subword_len, axis=1)

      # Repad the upsampled sequence if needed.
      if retiled_block_scores.shape[1] < length:
        repad_amt = length - retiled_block_scores.shape[1]
        repadding = tf.constant([[0, 0], [0, repad_amt], [0, 0]])
        retiled_seq = tf.pad(retiled_seq, repadding)
        retiled_block_scores = tf.pad(retiled_block_scores, repadding)

      # Make sure everything is the right length and add new dimension to concat
      # candidate blocks on.
      retiled_block_scores = retiled_block_scores[:, :length, :, None]
      retiled_seq = retiled_seq[:, :length, :, None]
      all_block_scores.append(retiled_block_scores)
      all_sequences.append(retiled_seq)

    block_scores = tf.concat(all_block_scores, axis=-1)
    block_scores = tf.nn.softmax(block_scores, axis=-1)
    candidates = tf.concat(all_sequences, axis=-1)

    # TODO: Block score calibration / block-by-block attention is omitted in this implementation.
    # batch_size x num_candidates x length x dim
    candidates = candidates * block_scores
    output = tf.reduce_sum(candidates, axis=-1) # bsz x length x dim

    # Downsample by mean pooling.
    if self.downsample_rate > 1:
      output = tf.nn.avg_pool(
          output, (self.downsample_rate,),
          strides=(self.downsample_rate,),
          padding="VALID")
    return output
```

