# OpenReview forum: "Charformer: Fast Character Transformers via Gradient-based Subword Tokenization"
_ICLR.cc/2022/Conference — ICLR 2022 Poster_

### Official Review · Reviewer_Md4g · 2021-10-28

**Correctness:** 3
**Technical Novelty And Significance:** 3
**Empirical Novelty And Significance:** 3
**Recommendation:** 6
**Confidence:** 4

**Main Review:**

Strengths:

1. The paper proposes a novel gradient-based tokenization module to address the issue of rigid subword tokenization algorithms and attains competitive performance. The proposed method saves engineering efforts for tokenization, which would be easy to generalize to different tasks.

2. It is interesting to find that this module can deal with multilingual and noisy datasets to some extent, which gives credit to learning subword representations from characters.

3. This proposed token-free module (GBST) has less computational cost than using the general methods of tokenization and can be well extended to other models.

Weaknesses:

1. The performance is not surprising, though the model indeed saves model size. It is not clear if the method works for other languages. For example, character-based models tend to achieve similar performance compared with word-based ones. Would this method has advantage over the character-based baseline then?

2. The evaluations are based on base models. Since this model saves model size, would it achieve better performance compared with the same size (e.g., 200M like Byte-level T5 Base in Table 1)? Do you try large models? Besides, it would be impactful if this paper showed that GBST module could actually generalize in other kinds of models.

Minor Comments:

1. I wonder if average pooling is good way to down sample though it is really fast, but how about MLP?

2. In Eq.3, the indices should be 0 to M-1 or 1 to M (total number of block size is M), i.e. $P_i = \softmax([p_{0, i},  p_{1, i},  \dots, p_{M,i}]$ -> $P_i = \softmax([p_{1, i},  p_{2, i},  \dots, p_{M,i}]$

3. At Figure 1 (b), it seems the annotation `P5:8` in the fourth line is missed.

4. In section 2.1.2, block sizes 1 < b < M should be 1 <= b <= M

5. In section 2.1.5, “latent subwords $\hat{X} = [\hat{X}_i, \dots, \hat{X}_M]$  should be “latent subwords $\hat{X} = [\hat{X}_i, \dots, \hat{X}_L]$


**Summary Of The Paper:**

This work aims to let the model learn the subword representations by itself from character-level embeddings of the input without tokenizing. A gradient-based subword tokenization module (GBST) is designed for replacement of the general tokenization. The proposed CHARFORMER, a re-scaled Transformer architecture (transformer with GBST) that integrates gradient-based subword tokenization, outperforms several byte-level baselines and performs on par with some subword-level models on three kinds of benchmarks, including GLUE, multilingual, and noisy text datasets. Moreover, CHARFORMER has the advantages of less memory usage, faster speed compared to the models with similar parameter size.

**Summary Of The Review:**

This work is well-motivated. The overall design of the model is technically sound, and it provides moderate improvements on several datasets, while being memory efficient, faster, with fewer parameters. It is not clear if the advance of the method can generalize to other languages or larger models.

---

> ### Author Response · Authors · 2021-11-17
> **Response to Reviewer Md4g**
>
> We thank you Reviewer Md4g for your feedback.
>
> > “The performance is not surprising, though the model indeed saves model size. It is not clear if the method works for other languages.”
>
> Our work presents the first pretrained character-level Transformer model that is comparable in performance to T5 while being competitive in speed to T5. We believe this efficiency gain coupled with the simplicity of the model represents a significant contribution over previous works (ByT5 which loses speed and CANINE which loses quality, generality, and simplicity), making pretrained token-free models much more useful for practitioners. In Section 3.3 and in Table 4 we describe our extensive multilingual experiments, which shows that our method does work well for a variety of languages against very strong baselines (mT5).
>
> >“Since this model saves model size, would it achieve better performance compared with the same size (e.g., 200M like Byte-level T5 Base in Table 1)? Do you try large models?”
>
> In Table 1 and 4 we report our comparisons between all models at the same size (200M Base as well as 134M SBase). Regarding trying large models - thank you for the suggestion, we are currently running a ~1.23B version of Charformer at the same parameterization as ByT5 Large, and we will update the submission with this result by the end of the rebuttal period as it finishes.
>
> >“ it would be impactful if this paper showed that GBST module could actually generalize in other kinds of models.”
>
> We show the performance of GBST in a standard Transformer backbone, adopted from T5. As Transformers are very widely used in the current NLP literature, we believe that our experiments capture the generality of the method well.
>
> >Minor Comments:
>
> - “I wonder if average pooling is good way to down sample though it is really fast, but how about MLP” While there are many ways to downsample that may be useful for GBST and may be worth future exploration, mean-pooling is indeed chosen as it is very efficient. It also has a nice property that if two neighboring positions choose the same block representation (via the block scoring network) - taking the mean of them losslessly merges them into one representation.
> - All other comments have been incorporated into the paper, thank you for the suggestions.

---

### Official Review · Reviewer_NyMb · 2021-11-01

**Correctness:** 4
**Technical Novelty And Significance:** 4
**Empirical Novelty And Significance:** 4
**Recommendation:** 8
**Confidence:** 5

**Main Review:**

Strengths:
+ The work makes tokenization more adaptable by learning sub-tokens on-the-fly.
+ Training speed and parameter efficiency are improved over more rigid, sub-word tokenization based Transformers in many cases.
+ The tokenization method is based on a weighting of pooled byte n-gram embeddings. Score calibration and downsampling are technically sensible and the authors make a noticeable effort to ease optimization and not introduce foreseeably brittle complexities like extra hyperparameters.
+ The n-gram weighting approach allows the sub-token weightings to remain somewhat interpretable.
+ The downsampling operation allows an easy way of saving later transformer stack parameter size, which is shown to increase learning speed and reduce parameters.
+ Potential limitations that require future work are mentioned proactively.
+ Instances of reasoning about technical choices, e.g. "narrower encoders", provide instructive details and help make the foci of the experiments more deducible.
+ Experiments are intentionally controlled (deconflated), to produce more robust insights.
+ Table 2 uses AUC-PR rather than the original works AUC-ROC as performance measure, which can be considered an improvement.

**Summary Of The Paper:**

This work first implements a byte-level (character) tokenizer, which is learned as part of the model rather than as a preprocessing task. To limit the computational burden of character level encoding on the downstream architecture, the method creates byte n-grams and combines them via a scoring network as outlined by the authors. Common Transformer architecture can use the tokenizer and even be narrowed. Finally the model is evaluated using multiple NLU and noisy NLP tasks. As a result, the approach achieves comparable end-task performance to sub-word and other byte-level Transformers, while improving parameter efficiency as well as ease of use for non English NLP tasks.

**Summary Of The Review:**

This work first implements an end-to-end trainable byte-level (character) tokenizer, while limiting computational overhead via a learned token weighting. It is compatible with common Transformer stacks, but can reduce their width for computational gains, without sacrificing performance, and in some places gaining performance. The evaluation on multiple NLU and noisy NLP tasks demonstrates only slighly less to slightly stronger performance compared to byte and sub-word tokenization models. Ease of use, being conceivably language agnostic, training more efficiently and allowing easy reimplementation (pytorch), make this work a welcome contribution to the field.

Therefore, I recommend accepting this work.

---

> ### Author Response · Authors · 2021-11-17
> **Response to Reviewer NyMb**
>
> We thank Reviewer NyMb for their feedback and review.

---

> > ### Comment · Reviewer_NyMb · 2021-11-30
> > **Response to authors**
> >
> > The additional experiments are interesting, especially the comparison to 1D convolutions - despite the inefficiency of the 1D convolution method. The works focus on reducing compute requirements is welcome, since concerns for medium sized LMs are often overlooked and help wide-spread application. The rebuttal requested scaling experiments are interesting, but should not become a standard requirement as this potentially creates inclusion issues downstream. Especially for methods where a core goal is increasing efficiency.
> >
> > The method itself performs well (on par or better with related methods). It allows for more dynamic tokenization while also balancing computational requirement that often prohibit the use of related methods. The authors provided additional convolution baseline experiments to demonstrate this advantage in quality and compute. Finally, it is easy to combine with existing models.
> >
> > Thus I will keep my initial score.

---

### Official Review · Reviewer_YTQy · 2021-11-02

**Correctness:** 3
**Technical Novelty And Significance:** 3
**Empirical Novelty And Significance:** 2
**Recommendation:** 6
**Confidence:** 4

**Main Review:**

Strength

1. The GBST module is simpler and faster. It has on-par performance but is 35%-80% faster with less memory usage than current SOTA byte-level transformer ByT5. Comparing with the lighter ByT5+CANINE model, it has similar speed and memory usage, but more straight forward and has better accuracy. It also has on par performance and less computation costs in most tasks comparing with T5/mT5.
2. The re-scaling of the model (SBase) achieves better performance with ~20% less computation. Though this idea to have deeper encoder is from previous work, it’s still informative to have the experiment.
3. The paper is very well written and has enough experiments to show the model’s performance in accuracy and capability in different tasks.

Weakness

1. (minor) The technical novelty of this paper is only GBST, a dense layer over n-grams, while training setting follows ByT5, and the best SBase model is also an implementation of the deep encoder concept.
2. The scalability of the model is not shown. All the experiment focusing on models (other than mT5) with a parameter scale of 200M. But the paper ByT5’s smallest model has 300M parameters and performs much better than with 200M parameters (comparing Table 4 of ByT5 and Table 4 of this paper).
3. (minor) T5 seems much better than SBase under Zero-Shot settings, though it takes more computation.
4. From the last sentence of 3.1 Baselines, it seems that the ByT5 model used for comparison is unscaled, and I couldn’t find in the paper about structure of this ByT5 (sorry if I missed that). This is important since in the ByT5 paper all the model variants have much heavier weight in encoders, and there is also an ablation study to prove that heavier encoder can improve performance a lot. In this paper, the scaled Charformer-SBase is also much better than the unscaled Charformer-Base. Thus it’s a bit strange that the best settings of both models are not used for comparison.



**Summary Of The Paper:**

This paper focus on a gradient-based subword tokenization (GBST) method for byte-level transformers. The key idea is to use a linear transformation (with trainable parameters) + softmax to compute embedding for each character based on several n-grams in its neighborhood, and then use average pooling to downsample and decrease the sequence length.

**Summary Of The Review:**

To summarize,

Pro

1. Excellent paper writing.
2. Extensive experiments.
3. Simple but effective token-free module.

Con

1. Lack of experiment on different sizes of the model, especially performance when with a larger scale.
2. The comparison seems not between the best setting (deep encoder, shallow decoder) of both baseline and this model.

---

> ### Author Response · Authors · 2021-11-17
> **Response to Reviewer YTQy**
>
> We thank Reviewer YTQy for the review. We respond to the reviewer’s comments below.
>
> >“Lack of experiment on different sizes of the model, especially performance when with a larger scale.”
>
> Thank you for the suggestion, we are currently running a ~1.23B version of Charformer at the same parameterization as ByT5 Large, and we will update the submission with this result by the end of the rebuttal period as it finishes.
>
> >"ByT5 model used for comparison is unscaled”
>
> At the end of Section 3.1 “Baselines” we mention that SBase results are only intended for comparison against Subword T5. Regardless, we have updated the results with an SBase scaled version of Byte-level T5. We observe that at this scaling, Charformer outperforms Byte-level T5.
>
> >"T5 seems much better than SBase under Zero-Shot settings, though it takes more computation.”
>
> Our evaluation agrees with previous results observed by Xue et al. [1], and only confirms the need for future works in this area. We believe that making character-level models simpler and more efficient is an important step towards this.
>
> [1] Linting Xue, Aditya Barua, Noah Constant, Rami Al-Rfou, Sharan Narang, Mihir Kale, Adam
> Roberts, and Colin Raffel. ByT5: Towards a token-free future with pre-trained byte-to-byte models.
> arXiv preprint arXiv:2105.13626, 2021. URL http://arxiv.org/abs/2105.13626.

---

### Official Review · Reviewer_zaHb · 2021-11-03

**Correctness:** 2
**Technical Novelty And Significance:** 2
**Empirical Novelty And Significance:** 2
**Recommendation:** 5
**Confidence:** 4

**Main Review:**

Strengths
1. Avoiding the reliance of NLP models on tokenizers is an important problem. Tokenizers influence the input structure and the design of the model. The input structure is not amenable to graceful modifications after the initial training which can be limiting.
2. The idea to calculate position representations based on a weighted average of subword blocks is interesting. This way the representation captures information from nearby n-grams.
3. The experimentation has some competitive results and focuses on both performance and speed aspects.

Weaknesses
1. My main concern is that the proposed method does not seem to be satisfying the criteria for a proper tokenization method.
     * First, it does not learn to segment the input to different chunks to be fed in the contextualizer but rather downsamples the sequence in a fixed way. The latent subword representations could be captured by combing multiple convolutions and a simple pooling function per position, which makes the method seem less novel.
     * Second, the soft selection at each position is performed over blocks of different size that correspond to that position and does not take into account global information e.g. by computing scores of all possible segmentations to maximize for the most likely one.
2. The current framing would suggest comparison to existing tokenization methods like BPE or recent  optimized ones to the downstream task [1,2] in a controlled setting (e.g. similar vocabulary size and contextualizer design), but this was not explored at all in the evaluation. In addition, there is no proper evaluation of the learned segmentations other than the visualization of the learned weights for a single example.
3. The results are not that good compared to pure character-level models on average which makes results less exciting. Also, one of the claimed benefits of the proposed method is that is faster is mainly due to the fixed downsampling which is not something new and applies to other character-level baselines.
4. Studies that focus on tokenization like the ones cited below evaluate on tasks where tokenization is important for reaching state-of-the-art performance like machine translation. It might be worth evaluating there.

Question:
* In section 8.2 about monolingual datasets, was the downsampling rate optimized only for the proposed model?

[1] https://aclanthology.org/2020.findings-emnlp.120.pdf

[2] https://arxiv.org/pdf/2012.15671.pdf


**Summary Of The Paper:**

This paper aims to remove the reliance of NLP models to external tokenizers. It proposes a gradient-based subword tokenization module that can be used in any neural model. The module scores predefined candidate blocks in a soft way for each position and then it weight averages them to obtain a mixture of subword representations. The resulting sequence is then downsampled in a fixed way to make processing easier. The proposed method integrated in a deep narrow transformer model performs on par and slightly better than character-level baselines on monolingual and multilingual classification respectively. In some settings it also outperforms strong subword-level baselines.

**Summary Of The Review:**

The paper focuses on an interesting problem and has many experiments with state-of-the-art results in certain settings. My main concerns are regarding its novelty, framing, and evaluation.  Even though some of the results are competitive, the experiments do not provide enough evidence that the learned  tokenization is crucial for achieving the results.

---

> ### Author Response · Authors · 2021-11-17
> **Response to Reviewer zaHb**
>
> We thank the reviewer for their response to our paper. We address the reviewer’s comments below.
>
> Weakness 1 & 2: Our primary goal of this work is to overcome the weaknesses present in traditional tokenization, where segmentations are discrete and static, and require a fixed vocabulary that occupies valuable memory and parameters (100K+ vocab size for multilingual models depending on the number of languages vs. a fixed 256 used in our models.) Thus, the goal of this work is not to generate an explicit segmentation and provide intermediate evaluations for such a segmentation. Instead, this work offers a way to emulate and replace segmentation as it is used in real world pretrained Transformer models - which is to make the sequence length computationally practical and improve model performance. In other words, we believe that the proper evaluation of the learned segmentations here is how well the segmentation works for downstream tasks, which we thoroughly compare against the current state of the art subword tokenization as used in T5/mT5 models.
>
> Weakness 3: Our work presents the first pretrained character-level Transformer model that is comparable in performance to T5 while being competitive in speed to T5. Even though downsampling was introduced previously by CANINE, we show that our method is both faster and more effective. Moreover, as discussed above, our method requires significantly fewer module parameters than previous works. Finally, in our updated SBase results, we observe that Charformer benefits from rescaling more than a strong character-level baseline (Byte-level T5), showing the additional value of this new inductive bias.
>
> Weakness 4: Although we do not evaluate on machine translation in this work, we do evaluate our models and all baselines on a very diverse set of tasks, domains, and languages. Particularly for multilingual task evaluations, having effective tokenization is crucial for task performance (see mT5 [1], which pays careful attention to vocabulary construction). Our work has extensive multilingual experiments and thorough comparisons to mT5.
>
> >”In section 8.2 about monolingual datasets, was the downsampling rate optimized only for the proposed model?”
>
> Local attention strided convolution (LASC, from CANINE) and GBST are the only two methods that have a downsampling rate parameter. For LASC we use the downsampling rate of 4 as introduced by the original work. All English GBST results use 2 or 3 tuned depending on the task, all multilingual results use 2. Additionally in Table 9 in the Appendix we show that GBST outperforms LASC at the same downsampling rate of 4 on TyDiQA-GoldP.
>
> [1] Linting Xue, Noah Constant, Adam Roberts, Mihir Kale, Rami Al-Rfou, Aditya Siddhant, Aditya Barua, Colin Raffel: mT5: A massively multilingual pre-trained text-to-text transformer. CoRR abs/2010.11934 (2020)

---

> > ### Comment · Reviewer_zaHb · 2021-11-30
> > **Response to authors**
> >
> > Thank you for the reply! I think that the current framing/conceptualization is what makes the reader expect detailed discussion and comparison with alternative segmentation baselines under controlled configurations. My concern is that the architectural choices that lead to the computational benefits are not new and the over-emphasis on state-of-the-art makes it difficult for the reader to discern what hypotheses are being tested exactly and what is the new thing that we learn from it. Therefore, I am inclined to maintain my initial recommendation.

---

### Official Review · Reviewer_yH31 · 2021-11-09

**Correctness:** 2
**Technical Novelty And Significance:** 3
**Empirical Novelty And Significance:** 3
**Recommendation:** 5
**Confidence:** 3

**Main Review:**

- (Unclear) How is the proposed GBST different from convolution, despite the scoring part?
- (Pro) Extensive evaluation results across many NLP classification tasks in mono- and multi-lingual settings.
- (Pro) Charformer seems to achieve similar predictive performance to Byte-level T5 while being more computationally efficient (Table 6).
- (Mixed) Charformer achieves worse predictive performance than subword models BERT and T5, which seems to suggest the proposed GBST gives little lift from character-level to subword-level. However, the Charformer models presented are smaller in size or faster in speed, or more efficient in FLOPS. Thus, hopefully, it will provide people with more choices for their use cases.
- (Mixed) Experimental results do not show strong improvements on accuracy metrics. Table 1-3 shows that Charformer is generally on par with Byte-level T5. Sometimes Charformer wins; other times Byte-level T5 wins. Either with small differences. In the multilingual case (Table 4), we do see Charformer consistently outperforms Byte-level T5 with a small margin. In all cases, Charformer is significantly worse than subword models. The rescaled version Charformer_S is able to achieve comparable and sometimes better accuracy than subword models. However, the re-scaling is not tied to Charformer and can be applied to other character-level models as well. It is not clear if the gain is because of the proposed GBST or solely due to the re-scaling.
- (Con) It is not clear if the proposed GBST module learns meaningful subword tokenization. Better qualitative evidence, e.g. examples that highly scored subwords align with human intuition, or quantitative evidence, e.g. how the learned tokenized subwords align with established methods, or how they are better at solving lexical tasks, would help.
- (Con) Many technical details are unclear/confusing. It would help if the authors can provide more and clearer technical details. If limited by pages, the author can point readers to appendices. Make sure the main text still contains the necessary details when moving things to Appendices.
- (Con) No code provided. Providing code will greatly help reproducibility and help clarify many technical details not fully described in the paper.

### Questions to the Authors
- How is the proposed GBST different from convolution, despite the scoring part?
- Can the authors help clarify the design of GBST and implementation of Charformer? For example, what is the Transfomer stack used by Charformer? It seems like T5, but I don't find it mentioned anywhere. See other localized points below.
- All the tasks in the paper seem to be classification tasks. Is Charformer encoder only? If so, how do the authors ensure a fair comparison with encoder-decoder models, e.g. T5? If not, is the decoder part of Charformer character-based, or it also uses some notion of subwords?

### Localized Points
- Eq (1). What is used as $F$ in the experiments?
- In "Considering Offsets". How is 1D convolution applied to X? How does it save computation?
- Figure 2. How to interpret this heatmap for block size > 1? How are they aligned? For example, we see a high score for block size = 3 at "k", is it for subword "tok" or subword "ken"?

**Summary Of The Paper:**

The authors propose a soft gradient-based subword tokenization (GBST) module with the aim of improving tokenizer-free end-to-end training of language models. The GBST module takes a byte-level sequence and computes all possible subword representation (up to a length) in a convolution-like manner, the results are then pooled, scored, and weighted with other add-ons including 1D-convolutions and an attention-like position-wise score. The authors use the GBST module followed by an encoder-decoder Transformer stack similar to that of T5 and call it Charformer. The authors conducted experiments on a range of NLP classification tasks under monolingual and multilingual settings. The trend seems consistent that subword-level models > Charformer ≈ byte-level T5 on monolingual and multilingual clean data, Charformer ≈ byte-level T5 > subword-level T5 for monolingual noisy data. In terms of efficiency, the authors show evidence that Charformer uses fewer parameters and FLOPS than subword T5 and proceeds more steps per second than byte-level T5.

**Summary Of The Review:**

A plausible proposal for the promising direction of learning subword tokenization end-to-end. However, some important design and implementation details are missing or confusing which are not helped by the lack of source code. The extensive experimental results themselves would benefit the community. However, they don't seem to strongly support the advantage of the proposed module in learning subword representation either in terms of end-task accuracy or explainability. If the authors could clarify what they did exactly with their GBST module and thus their novelty, I would be inclined to accept for the merit of extensive evaluation results and one more alternative that lies between subword-level and plain-byte-level models.

---

> ### Author Response · Authors · 2021-11-17
> **Response to Reviewer yH31**
>
> Thank you Reviewer yH31 for your detailed review. To provide more clarity on the technical details and implementation of GBST, we have updated the submission to include an implementation of the module in the appendix. Please let us know if you have any other questions regarding technical details. We respond to various points from the review below:
>
> >“How is the proposed GBST different from convolution, despite the scoring part?”
>
> A major difference of GBST over previous works is that it is very lightweight: using only a single convolution and shared linear transformation used for the block scoring network. We show that by non-parametrically pooling at various granularities and using a lightweight scoring network, we are able to outperform significantly heavier parameterization such as local attention and strided convolution [1]. Prior works such as [2] also required very large stacks of convolutions (upwards 1,600 filters) to achieve results comparable with explicit tokenization models. Here we show that we can achieve similar performance to subword T5 using a single convolution and a block scoring network. Finally, we have updated the paper to include a convolution-only ablation of GBST, to show the value of the block scoring and selection on performance.
>
> >“[The results] don't seem to strongly support the advantage of the proposed module in learning subword representation either in terms of end-task accuracy or explainability.”
>
> As pointed out by other reviewers, the contribution of Charformer beyond accuracy and explainability is its efficiency. Currently, the highest performing pretrained token-free models (e.g. ByT5) are too expensive for many practitioners to use, and more efficient ones such as CANINE (which we provide a comparison for in this paper) does not provide comparable results across general tasks to subword models. This paper provides a method of getting results comparable to subword models, using a model that is simpler and faster than previous methods. The resulting model is not only faster than previous token-free models but it is also competitive with subword mT5 (see Table 5.)
>
> >“All the tasks in the paper seem to be classification tasks. Is Charformer encoder only?"
>
> Charformer is an encoder-decoder model, and in this paper we do run the model on TydiQA-GoldP, XQuAD, and MLQA which have sequence outputs (the answer span). GBST is applied only on the encoder side (right before the encoder Transformer stack), and the decoder decodes individual bytes. The application of GBST for the decoder however is an interest for future work.
>
> >“It is not clear if the proposed GBST module learns meaningful subword tokenization.”
>
> In the paper we provide a visualization of how GBST scores n-grams at various positions for an example sequence, and while we believe this visualization can provide some insight on the GBST process, we also would like to highlight a main feature of GBST is its soft nature. The model is encouraged to combine local information in a way best for it to solve the task. Importantly, this may or may not align with human intuition or hand designed segmentation algorithms. In this sense, instead of evaluating the intermediate segmentations produced by the model, we evaluate the whole model end-to-end as how the tokenization affects final model performance. In the end, GBST accomplishes the same end goal as traditional tokenization: reducing the sequence length (efficiency) while enabling downstream performance (comparison with the T5 baseline).
>
> >Localized points
>
> - “For example, what is the Transformer stack used by Charformer?” Charformer uses the T5 stack with the addition of the GBST module written in Mesh Tensorflow.
> - “Eq (1). What is used as F?” We use mean pooling.
> - “How is 1D convolution applied to X? How does it save computation?” Using just offsets would require enumerating and scoring all overlapping blocks sized 1 to 4. Instead we apply a single 1D convolution across the sequence length and only form non-overlapping blocks of size 1 to 4, which saves computation (much fewer blocks to score).
> - “How to interpret this heatmap for block size > 1? How are they aligned?” Here we have to keep in mind that a 1D convolution of size 5 was applied before forming non-overlapping blocks of sizes 1 to 4. Thus, the block-3 score for “k” could potentially represent its membership in both “tok” and “ken” (any possible block 3 membership).
>
> [1] Clark, J. H., Garrette, D., Turc, I., & Wieting, J. (2021). Canine: Pre-training an Efficient Tokenization-Free Encoder for Language Representation. ArXiv Preprint ArXiv:2103.06874.
>
> [2] Lee, Jason, Kyunghyun Cho, and Thomas Hofmann. "Fully character-level neural machine translation without explicit segmentation." Transactions of the Association for Computational Linguistics 5 (2017): 365-378.

---

> > ### Comment · Reviewer_yH31 · 2021-12-04
> > **Response to authors**
> >
> > Thanks for the reply! I think it would be great to incorporate the discussions/explanations here into the paper.
> >
> > The lack of discussions/comparisons for learned subword tokenizations remains a major concern. The current framing emphasizes the end-to-end learning of tokenization (even the title features tokenization). One would expect to see solid discussions on how and how well the method produces sensible tokenizations. Echoing Reviewer zaHb, adding discussions or changing the framing would help.
> >
> > I'm not convinced by the authors' claim that their proposal could achieve comparable performance with subword-level T5 while being more efficient. The results I see suggest subword-level models > Charformer ≈ byte-level T5 in performance and subword-level models ≈ Charformer > byte-level T5 in efficiency. It is a good thing we improve efficiency while also slightly improving predictive performance compared to byte-level models, but I think it would be an overclaim that Charformer performs comparably to subword-level models.
> >
> > Besides, much of the gain seems to come from the down-sampling (SBase vs Base), which is also applicable to other byte-level baselines.
> >
> > I do feel the extensive evaluations could benefit the community and the paper provides an alternative model design for the performance-efficiency trade-off. However, I remain concerned about the novelty and validity of the key technical contribution, the GBST module. It is not properly compared as a tokenization method and its effectiveness is not convincingly demonstrated throughout the downstream tasks. I thus remain my initial recommendation.

---

### Author Response · Authors · 2021-11-17
**Overall Response & Paper Revision**

Thank you all reviewers for your time and effort in responding to our work. We really appreciate the feedback. In response to your comments we would like to bring your attention to the following additions in the revision of the paper:

- For additional technical clarity, we have included an example Tensorflow implementation of GBST in the appendix.
- We have included a convolution-only ablation baseline in comparison to GBST in Table 1. This result compares a 1D convolution model to GBST (essentially GBST without the block scoring, block ranking, and downsampling). The 1D convolution here used a filter size of 5, which is the same as the one used in the GBST model. As there is no downsampling, this model is also more expensive than GBST. GBST outperforms this baseline in most evaluated tasks. We hope this can shed more empirical insight on the contribution of the block scoring network and score based soft selection over only convolutions.
- We have included an additional comparison of Byte-level T5 with the SBase scaling. Given that this model is pretrained with the same experimental settings as Charformer SBase, the results show that the GBST inductive bias has stronger performance under this scaling than vanilla Byte-level T5.
- We are currently running an additional scaling experiment for Charformer where we scale Charformer using the same parameterization as ByT5 Large (~1.23B params). We will update the revision of the paper with these results before the end of the rebuttal period.

---

> ### Author Response · Authors · 2021-11-22
> **Revision update with scaling results**
>
> We have now uploaded a new revision of the paper with an added “Large-scale Experiments” section in the Appendix (Section 8.5) containing our preliminary results for Charformer at the same scale as ByT5-Large (~1.23B). When considering the new results however, please note that due to rebuttal period time constraints, the Charformer model is only pre-trained to ~900K steps as opposed to 1M steps. Even then, we see that when scaled up the Charformer model outperforms mT5 with the same number of parameters, while being ~2x faster than ByT5.

---

### Decision · Program_Chairs · 2022-01-20

**Decision:**

Accept (Poster)

**Comment:**

This paper introduces a soft gradient-based subword tokenization module (GBST) that learns latent subword representations from characters. GBST enumerates candidate subword blocks and learns to score them in a position-wise fashion using a block scoring network. The resulting model was tested on GLUE, and several cross-lingual tasks. The performance is competitive with ByteT5 and often similar to subword models while being more efficient in FLOPs and RAM.

Reviewers are mixed on this. The negative reviewer points to how this not being a real tokenizer and does not produce a tokenization, that experiments that use the base model do not address bigger scales, and that there is a lack of code which is important for this kind of work, and the resulting accuracy gains are not significant and the method being not interpretable. The positive reviewers like the extensive experiments, the efficiency improvements and flexibility / simplicity of the GBST module. The authors seemed to have addressed most of the reviewer issues by providing larger scale experiments and code. I believe the results are fairly strong, since one would not expect a big performance difference in a learned tokenization method, but rather efficiency or flexibility gains. The paper is generally well-written though details about the convolution should be included in the text (and not just the code).

Recommending accept.